# The Expansion of the Hellenic Food Thesaurus; Allergens Labelling and Allergens-Free Claims on Greek Branded Food Products

**DOI:** 10.3390/nu14163421

**Published:** 2022-08-19

**Authors:** Alexandra Katidi, Antonis Vlassopoulos, Stefania Xanthopoulou, Barbara Boutopoulou, Dafni Moriki, Olympia Sardeli, José Ángel Rufián-Henares, Konstantinos Douros, Maria Kapsokefalou

**Affiliations:** 1Department of Food Science & Human Nutrition, Agricultural University of Athens, 11855 Athens, Greece; 2Department of Nursing, National and Kapodistrian University of Athens, 11527 Athens, Greece; 3Allergology and Pulmonology Unit, 3rd Pediatric Department, National and Kapodistrian University of Athens, 12462 Athens, Greece; 4Departamento de Nutrición y Bromatología, Instituto de Nutrición y Tecnología de Alimentos, Centro de Investigación Biomédica, Universidad de Granada, 18071 Granada, Spain; 5Instituto de Investigación Biosanitaria ibs.GRANADA, Universidad de Granada, 18071 Granada, Spain

**Keywords:** branded food database, allergen, ingredient list, precautionary statement, allergen-free claims, HelTH, Greece

## Abstract

Branded food composition databases (BFCDs) are valuable information tools that meet multiple user needs. Recently, recognising allergies and intolerances as an emerging concern for various stakeholders, BFCDs evolve to embed information on allergens. This study aims to expand the Greek BFCD, HelTH, to include allergen information for its 4002 products. A new file was added to the structure of HelTH, and data were curated to record label information. In 68.4% of products, at least one allergen was present in the ingredient list and in 38.9% at least one allergen in a precautionary statement. Milk (38.8%), gluten (32.7%), and soybeans (17.4%) were most commonly declared in the ingredient list; nuts (18.3%), eggs (13.1%), and milk (12.2%) were most commonly declared in precautionary statements. Allergen-free claims were present in 5.3% of the products and referred mostly on gluten and milk. In general, no statistically significant differences were identified between the nutritional composition of allergen-free claimed products and their equivalents. This study delivers an expanded BFCD that provides organised and detailed allergen information; new insights on the presence of food allergens in branded foods and issues of concern regarding allergen declaration that need to be addressed in order to improve label information.

## 1. Introduction

Branded food composition databases (BFCDs) have been developed over the years to provide data on the nutrient composition of branded foods, some also on other important features they may carry, such as nutrition and health claims and quality indicators [1,2,3,4,5,6,7]. BFCDs are becoming a powerful and valuable information tool for research, new product development, dietary advice, food and nutrition policy, and food business. Consumers may also refer to BFCDs, mostly through digital diet applications that embed BFCDs, to make informed choices that adapt their dietary intake [8]. It follows that BFCDs must constantly expand and evolve their features to reflect market advances and satisfy a variety of emerging needs of users.

Health concerns remain a trend in food product development [9]. This includes not only lifestyle-related diseases (e.g., diabetes, obesity, and hypertension) [10,11], but also allergies and intolerances [12,13,14,15,16,17,18]. In recent years, consumers are seeking out foods and beverages that protect them against real or perceived allergens; thus, they search the ingredient list and all relevant allergen information on the label to avoid unwanted ingredients [19,20]. In addition, gluten-free eating patterns have become a mainstream phenomenon during recent years, with perceived healthiness being the fundamental reason for this choice [21,22,23]. Moreover, consumer research shows that lactose-tolerant or undiagnosed population groups actively search for lactose-free products that they link with several potential health-related benefits [24,25].

Labels are clearly the means to inform on the presence or potential presence or absence of allergens, thus must be clear, specific, accurate, and understandable. Labelling regulations have been developed by the international legislation bodies [14,26]. The European legislation selects 14 food ingredients as food allergens and provides mandatory general guidelines for the allergen labelling of packaged foods [27]. The 14 allergens are cereals containing gluten (wheat, rye, barley, spelt, kamut, or their hybridized strains), crustaceans, eggs, fish, peanuts, soybeans (including soya lecithin), milk (including lactose), nuts (namely almonds, hazelnuts, walnuts, cashews, pecan nuts, Brazil nuts, pistachio nuts, macadamia, or Queensland nuts), celery, mustard, sesame seeds, sulphites, lupin, and molluscs, including products thereof. Allergens are declared on pack as part of the ingredient list in a manner that distinguishes them from the rest of the ingredients, such as a bold typeset, text highlights, or positioning the end of the ingredient list preceded by a statement such as “contain” [12]. Moreover, precautionary labels warn on adventitious presence of allergens (unintentional contamination by contact with other products during processing, storage, or shipping) [28,29,30]. It follows that, despite efforts, allergen declaration remains complicated [31,32] and the need to organise and document allergen information on the presence of allergens emerges. The BFCDs have been recognised as appropriate information tools to embed allergen data by various national organisations (e.g., OQALI, USDA) or international initiatives (e.g., Stance4Health-S4H) [12,15,33,34]. Thus, in the evolution strategy of BFCDs, the introduction of information on allergens is a reasonable and appropriate decision.

In Greece, the Hellenic Food Thesaurus (HelTH) is the Greek BFCD; it presents data on the nutritional composition of foods, any health and/or nutrition claims, other quality indicators written on pack (environmental claims, origin, etc.), and graphical indicators (logos) [5]. HelTH was launched in 2019 by the Agricultural University of Athens and currently includes *n* = 4002 [3]. Recognising allergens as an emerging issue of interest for users, the expansion of HelTH to provide information on allergens emerges as a priority.

This study aims to (i) expand the HelTH BFCD, (ii) record the overall presence of allergens in the branded foodscape in Greece, (iii) record the presence of allergens in the ingredient list and in a precautionary statement, and (iv) describe the presence of allergen-free claims and compare the nutritional composition of products carrying allergen-free claim to their corresponding counterparts.

## 2. Materials and Methods

### 2.1. Expanding the HelTH Structure

To register information on allergen presence, the structure of the BFCD HelTH was expanded to include a new file, named “allergens’ file” (Figure 1). The file, tracked information on any of the 14 allergens mentioned at the EU regulation [28]; these are cereals containing gluten (wheat, rye, barley, spelt, kamut, or their hybridized strains), crustaceans, eggs, fish, peanuts, soybeans (including soya lecithin), milk (including lactose), nuts (namely almonds, hazelnuts, walnuts, cashews, pecan nuts, Brazil nuts, pistachio nuts, macadamia or Queensland nuts), celery, mustard, sesame seeds, sulphites, lupin, and molluscs, including products thereof.

In total, 42 new variables were introduced in the allergens’ file; namely, for each of the 14 allergens, (a) a variable “x allergen used as ingredient”, (b) a variable “x allergen used in a precautionary statement”, and (c) a variable on the presence of a “free-from x” claim. Summary variables were created to describe for each food (i) the total number of allergens in the ingredients list, (ii) the total number of allergens in a precautionary statement, and (iii) the total number of allergen-free claims.

### 2.2. Curating Allergen Data and Allergen-Related Claims in the Expanded HelTH Database

Labelling information on allergens was searched on the food labels of the HelTH branded food products (*n* = 4002) from July 2021 to October 2021 and subsequently entered into the expanded structure of the HelTH database according to the following procedures/criteria:(A)For recording the presence of allergens in a given food in the variable “allergens used as ingredients”: an ingredient should be among the 14 allergens listed in the European Regulation [28] named and highlighted in the ingredient list or at the end of the list preceded by statement such as “contain”. Presence of milk allergens in the “traditional yogurts” subcategory and in some cheeses was recorded regardless of presence of ingredients list according to Regulation [35].(B)For recording the potential presence of allergens in a given food in the variable “allergens used in a precautionary statement”: an ingredient should be among the 14 allergens listed in the European regulation [28] and mentioned on the label with a statement such as “may contain traces of”, “manufactured in a facility that also processed”, “may be present”. Adventitious presence and presence of traces were grouped.(C)For recording “allergen-free” claims, all sides of the packaging were checked to identify any on-pack communication, including logos [22], indicating the absence of any of the 14 allergens listed in the European Regulation [28].

Allergen data entered at the HelTH database were double-checked for accuracy by two independent researchers from October 2021 to March 2022.

### 2.3. Analysis of Prevalence of Allergens in the Greek Branded Foodscape

Every branded food product that mentioned at least one of the 14 allergens in the ingredient list and/or in a precautionary statement was considered a product containing allergens. An allergen declared both in the ingredient list and in a precautionary statement was accounted as part of the ingredient list only.

In accordance with the regulation, some ingredients or substances derived from listed allergens were not considered as allergens (for example wheat-based maltodextrins or fully refined soybean oil and fat) [35].

### 2.4. Comparison of the Nutritional Composition of Products Carrying Allergen-Free Claims to Their Corresponding Counterparts

Energy and macronutrients of interest (protein, total fat, saturated fatty acids (SFA), total sugars, and salt) were checked to find out similarities and differences between the nutritional composition of products in a specific subcategory carrying allergen-free claims and similar products in the same subcategory that do not. Comparisons were carried out in food subcategories that included at least 5 products with an allergen-free claim and 5 without one. Due to their large diversity, the food subcategories “Spices or condiments” and “Non-alcoholic beverages” were excluded from these analyses; these constitute of products that are very different, e.g., spice cubes and ketchup in “Spices or condiments” and cocoa powder and soft drinks in “Non-alcoholic beverages”. Thus, the results of the nutritional comparison between very different products bearing allergen-free claims and the ones that do not, could be incorrectly interpreted and/or misleading. Statistical analysis was carried out using IBM SPSS Statistics^®^ (version 23, Northridge, CA, USA). Nutritional composition data were analyzed as continuous variables (content per 100 g or 100 mL of product). Data were tested for normality using the Kolmogorov–Smirnov test. None of the variables followed the normal distribution. Therefore, variables were expressed as median (interquartile range). Differences were tested using the Mann–Whitney U non-parametric test for 2 independent samples. Statistical significance was set at 0.01% to adjust for multiple comparisons (Bonferroni correction).

## 3. Results

### 3.1. Introducing Allergen Data in the Expanded HelTH Database

In total, 3859 out of 4002 products in the HelTH BFCD declared on their labels information related to allergens (presence or absence); these are presented in Table 1 per food category and subcategory. In particular, out of the 4002 products that populate HelTH, 3915 products provided readable label data to allow for allergen screening. Of those, 56 food products (1.4%) did not comply with the legislation regarding the declaration of the presence of allergens on the label and were excluded as low-quality data.

The majority of the products either declared allergens in the ingredient list or as a potential cross-contamination, which was recorded in a precautionary statement (Table 2). Allergens were more commonly present in the ingredient list than in precautionary statements; 68.4% (*n* = 2640) of the food products declared at least one allergen in the ingredient list while 38.9% (*n* = 1501) had a precautionary statement for at least one allergen. Overall, 27.3% (*n* = 1052) of food products did not mention any allergen, neither in the ingredient list nor in a precautionary statement.

### 3.2. Allergens in the Ingredient List of the Food

Allergens most common in the ingredient list are presented per food subcategory in Table 3. These were milk (39%), gluten (33%) and soybean (17%), molluscs (0.3%), crustaceans (0.1%), and lupin (0.1%) are the least mentioned allergens.

Milk was present as an ingredient in 95% of “Savoury cereal dishes”, 77% of “Chocolates”, and 70% of the “Fine bakery wares” (dairy products and butter, were excluded from this analysis). Cereals, including gluten, were found in 100% of “Savoury cereal dishes”, 99% of “Fine bakery wares”, 97% of pasta or similar product” and “Bread or similar product” subcategories, 92% of “Breakfast cereals”, and 81% of the “Meat dishes”. Soybeans were found in 88% of the “Chocolate” subcategory, 59% of the “Fine bakery wares”, and 54% of “Frozen dairy desserts”, while eggs were mainly declared at “Ready-to-eat meals” (39%), “Savoury cereal dishes” (36%), and “Fine bakery wares” (31%).

From the less common allergens (Appendix A), fish, crustaceans, and molluscs were mainly found in the “Seafood or seafood product” category; similarly, lupin was only found in “Grain or grain product”. On the contrary, sulphites, although declared in only 2.7% of HelTH, were highly common in subcategories such as “Ready-to-eat meals” (45%), “Processed fruits” (35%), and “Starchy roots or potatoes” (24%) (Appendix A). Celery was found in ~20% of “Cereal or cereal milling product” and “Ready-to-eat meals” and to a lesser degree (~10%) in “Canned meats”, “Meat dishes”, “Prepared foods”, and “Spices or condiments”. As expected, nuts were declared predominantly in the “Nuts” subcategory (61%), with similar findings for sesame declared in 78% of “Nuts” and 56% in “Non-chocolate sugary products”. Mustard was mostly found in “Ready-to-eat meals” (32%), “Spices and condiments” (21%), “Prepared foods” (15%), and “Sausages or similar product”, “Canned meats”, and “Meat dishes” (~10% in each).

### 3.3. Allergens in a Precautionary Statement

Allergens most commonly declared in a precautionary statement are presented per food subcategory in Table 4. These were nuts (18%), eggs (13%), and milk (12%), followed by soybeans (11%), sesame (11%), cereals (9%), and peanuts (9%). Molluscs are the least declared allergen in a precautionary statement.

Nuts appeared in a precautionary statement in 63% of the “Fine bakery wares”, 58% of the “Chocolate” and the “Breakfast cereals” subcategories, 53% of “Canned meat”, 48% of “Frozen dairy desserts”, and 46% of “Non-chocolate sugary products”. Eggs appeared in a precautionary statement in 55% of the “Cereal or cereal milling products” subcategory, in 41% of the “Savoury cereal dishes”, and in 40% of the “Fine bakery wares”. Milk appeared in a precautionary statement in 49% of the “Cereal or cereal-milling product”, 41% of the “Bread or similar product” subcategories, and 38% of the “Meat dishes”.

In addition, the food subcategories with the highest use of a precautionary statement (at least one allergen) were “Nuts” (94%), “Fine bakery wares” (91%), “Ready-to-eat meals” (87%), “Savoury cereal dishes” (86%), “Chocolates” (83%), and “Meat dishes” (81%) (Appendix A).

In the “Nuts” food subcategory, the allergens most used in a precautionary statement, were sesame (65%), peanuts (61%), and cereals including gluten (58%). In the “Fine bakery wares” food subcategory, the allergens most used in a precautionary statement were nuts (63%) and eggs (40%). In “Ready-to-eat meals” were celery (63%), fish (37%), mustard (29%), and soybeans (29%). In “Savoury cereal dishes” were sesame (71%), mustard (44%), nuts (42%), and eggs (41%). In the “Chocolate” subcategory were nuts (58%), cereals (54%), and peanuts (32%), and in the “Meat dishes”, mustard (43%) and milk (38%).

From the less common allergens declared in a precautionary label, lupin was only found in the “Grain or grain products” category and “Prepared foods” subcategory with prevalences ≤8%. Crustaceans and molluscs were declared almost exclusively in “Ready-to-eat meals” (13% and 26%, respectively). Fish was declared in a precautionary statement in “Ready-to-eat meals” (37%) and “Prepared foods” (20%). Sulphites were declared as a precaution in “Canned meats” and “Sausages or similar products” (34% and 29%, respectively) followed by “Ready-to-eat meals” (16%), “Frozen semi-ready meals” (13%), “Nuts or seeds products” (11%), and “Sugar, syrup, or honey” (9%). Finally, mustard and celery, although present in the precautionary statement of ~6% of all foods, were listed in the 30–63% of foods in the “Meat and meat products” category, the “Ready-to-eat meals” subcategory, and even in the “Savoury cereal dishes” subcategory (Appendix A).

### 3.4. Allergen-Free Claims

The prevalence of the three most commonly used allergen-free claims is presented per food subcategory in Table 5. Overall, 5.3% (*n* = 206) of the 3859 products analysed carry an allergen-free claim. The following allergens were mentioned in allergen-free claims: gluten, milk, soybeans, sesame, nuts, eggs, celery, and mustard. The most used allergen-free claim was gluten-free (4.6%), followed by milk-free (1.6%), and soy-free (0.2%) claims. The absence of other allergens was rarely communicated on-pack. Out of the 38 food subcategories, 23 (60.5%) had at least one allergen-free claim (Appendix A). The “Milk imitation products” subcategory was the food subcategory with the highest use of allergen-free claims (71% of the products), and the only food subcategory in which all the allergen-free claims identified in the 3859 branded food products, were present on its products. Specifically, 71% of the “Milk imitation products” bear a milk (or milk products)-free claim, 47% bear a gluten-free claim, 10% bear a soy-free claim, 4% a nuts-free claim, and 2% (*n* = 1) a sesame, eggs-, mustard-, and celery-free claim (this product carries spontaneously all the eight allergen-free claims identified in the study). One product also carrying a sesame-free claim was found at the “Prepared food products” subcategory (0.6%), and another one carrying an eggs-free claim was found at “Breads or similar products”.

The allergen-free claims, identified in subcategories with at least five products bearing the claim and at least five similar products that do not, were “gluten-free” and “milk-free”. Therefore, comparisons on the nutritional composition of products carrying the allergen-free claim to similar ones that do not, could be carried out only for these two claims.

For the “gluten-free” claim, the food subcategories that meet the criteria and were included in the analysis are the following: yogurt, imitation milk products, preserved meat, sausage or similar meat, pasta or similar product, non-chocolate confectionary, and ready-to-eat meals. For the “milk-free” claims, only the milk and imitation milk products included at least five products with and five without a milk-free claim.

Figure 2 presents the comparison between products bearing a gluten-free claim and their corresponding ones, per macronutrient and per food subcategory. In general, products carrying a gluten-free claim do not differ from their corresponding ones. Statistically significant differences can only be found in the preserved meat subcategory for energy and total fat. In the preserved meat subcategory, products bearing a gluten-free claim seem to have approximately half the energy content and less than a third of the fat content than their corresponding ones.

Related to the milk-free claims, products bearing them do not present any statistically significant differences neither in the milk subcategory, nor in the milk imitations one (data not shown). However, when we compared products used as milk (the two subcategories grouped together) with and without a milk-free claim (milk-free claims include the lactose-free claims), then statistically significant differences were found. Specifically, products carrying a milk-free claim have a lower content of protein, saturated fatty acids, and total sugars compared to the corresponding products that do not carry this type of claim (data not shown).

## 4. Discussion

The most important outcome of the present study is the detailed information and the analyses on the prevalence of allergens in the ingredient list and/or in the precautionary statements and of allergen-free claims in branded foods in the Greek market. Observations on the prevalence of specific allergens in branded foods, the nutritional composition of products bearing allergen-free claims, as well as issues of concern associated with allergen declaration, enlighten the topic of allergens in the foodscape, which concerns an increasing number of consumers. This is the first study that provides information on allergens in foods in Greece.

Another important outcome is the expansion of the HelTH branded food database; the revised structure and methodology developed for allergen data curation delivered the expanded HelTH, which now embeds label information on the presence or absence of allergens in all categories and subcategories of the database, in an accurate and organised manner.

In specifics, the prevalence of allergens in branded foods was rather high: 68% (*n =* 2640) of the 3859 branded foods studied contained at least one allergen and 39% (*n =* 1501) declared at least one allergen in a precautionary statement. These results agree with those of a study conducted in France, in which the OQALI (the French Observatory of Food Quality) found that 73% of the 17,309 branded foods studied contained at least one allergen in their ingredient list while 39% had a precautionary statement for one or more allergens [12]. In another study, conducted in Latin America, 63.3% of the 10,254 branded foods studied declared at least one allergen while 33.2% declared allergens in a precautionary statement [15]. In contrast, in a smaller study conducted in 2011 in supermarkets of Melbourne, Australia 882 products (65%) of the 1355 branded food products observed, had a precautionary statement for one or more allergens [30].

The most common allergens in the ingredient lists were milk, gluten, and soybeans (Table 2), particularly due to the use of soya lecithin in the food formulation (data not shown). Similar results were observed in the OQALI study, the Latin America, and the Australian study, all declaring milk, gluten, soybeans, and eggs in similar frequencies to ours [12,15,30]. The most common allergens in a precautionary statement (Table 4) were nuts, eggs, milk, soybeans, and sesame. Similar results were observed in the OQALI study, where the most common allergens in precautionary statements were nuts, eggs, peanuts, soybeans, and milk [12]. In the study reflecting the food markets in Latin America, the most common food allergens declared at the precautionary statement were nuts, soybeans, and milk [15], while in Australia, the most common allergens listed at precautionary statements were nuts and peanuts, followed by sesame and eggs [30]. All studies mentioned above show that an important percentage of the branded food products declare allergens in a precautionary statement.

The frequency of products carrying allergen-free claims such as gluten-free, dairy-free, etc., is rather low; 5.4%. In the claims, the wording of the allergen-free claim was variable; “x allergen-free”, without “x allergen”, “free from x allergen” were most commonly used while in the milk-free claims, “lactose-free”, “dairy-free”, “casein-free”, “milk derivatives’-free” were also used. It should be mentioned that allergen-free claims and particularly, gluten-free, and dairy-free claims, were sometimes found as logos on products’ packaging (*n* = 74; 35.9%). Per allergen category, frequencies were between 0.03% (*n* = 1) for mustard and celery and 4.6% (*n* = 178) for gluten. Overall, eight allergens were mentioned at allergen-free claims (gluten, milk, soybeans, sesame, nuts, eggs, mustard, and celery). In comparison, the OQALI study identified five allergens of allergen-free claims, namely gluten, milk, peanuts, eggs, and soybeans in similar frequencies [12].

While recording the presence of allergens in the branded foods, issues regarding allergen declaration were identified. One issue was the absence of a uniform manner of presenting the declaration of allergens in the ingredient list or in the precautionary statement. For example, no standard typeset is adopted; allergens in the ingredients list may be declared in bold type letters or capital letters or both bold and capital letters and/or in a coloured background; allergens in a precautionary statement could use phrases such as “may contain traces of”, “may contain”, and “produced in a facility that processes…”; precautionary statements may not always appear after the ingredients list. Another issue was that substances were declared as allergens in some products but not in others. Eleven substances, namely lactase, lactic acid crops, glucose and/or fructose syrup, corn-starch, monosodium glutamate, acid sulphite caramel, barm (wheat), wholemeal flour, black sesame, soybean oil, and butter aroma were highlighted as allergens in some but not all branded food products. In particular, nine dairy products highlight the microbial cultures as allergens, one meat product highlights the monosodium glutamate as an allergen, three meat products the dextrose, one meat product the corn starch, one mixture for sweets highlights the antioxidant extract rich in soy tocopherols, one toast bread the black sesame, two fat products the butter aroma, and one fat product the milk aroma. However, on these issues, regulation (EU) No 1169/2011 [35], is specific (namely, glucose and fructose syrup and black sesame, are not considered allergens), although sometimes complicated. For example, soybean oil is not an allergen when fully refined; however, in an ingredient, list the degree of refinement is rarely mentioned leading to some products highlighting soybean oil as an allergen and others not, without an explicit explanation. Another issue specific to imported branded foods was also identified. According to legislation, all imported foods need to provide the ingredient list translated, which is often printed and pasted on top of the packaging. In this study we observed that in stickers with ingredients’ translation, wrong ingredients were often highlighted as allergens. During the data entry process, two meat products and one dairy product were identified as highlighting wrong ingredients as allergens, while on the original ingredient list, allergens were correctly highlighted. All these observed issues suggest that the consumer may be confused and/or not able to identify instantly the crucial information on the presence of allergens; thus, they highlight gaps in the legislation or its adoption that need to be addressed in the future.

For some consumers, it may be relevant to know not just the presence of an allergen but also the quantity of an allergen present. It must be noted that an on-pack allergen declaration does not inform on the quantity of the allergens present in the food. In most cases, quantities of allergens were not mentioned. In some food products, quantity information is provided for some ingredients but not all; those in small quantities, allergens included in most cases, are not mentioned. However, there is not a threshold or a standard way that these cut-offs are chosen. Thus, it appears that there was no uniform way that allergen quantities are presented in the ingredient lists. According to the Regulation (EU) No 1169/2011 [35], the indication of the quantity of an ingredient or category of ingredients used in the manufacture or preparation of a food shall be required where the ingredient or category of ingredients concerned: (a) appears in the name of the food or is usually associated with that name by the consumer; (b) is emphasised on the labelling in words, pictures or graphics; or (c) is essential to characterise a food and to distinguish it from products with which it might be confused because of its name or appearance. Thus, allergen exposure cannot be calculated from information on food labels.

In this study, limitations may be identified. Data were collected from label information, but label information may not be complete; undeclared allergens on food labels continue—year over year—to be the leading cause of food recalls in the United States [36]. This could lead to underreporting on the presence of allergens in branded foods. Limitations are also linked to the regulation framework; as mentioned in Europe, the wording is not regulated, threshold values concerning the smallest dose for an allergic reaction do not exist, and there are no limit values to establish if the allergen must be mentioned in precautionary labels [12,37]. This poses an extra burden in the curation and standardization of any attempt to systematically map the presence of allergens based on food labels [15]. In this study, labels were retrieved from retailers’ websites, which sometimes provide images of the product packaging not updated [2] or of poor quality, e.g., not all surfaces or unclear text [3]; however, as the results show, this problem, although identified, was very limited (56 products only).

## 5. Conclusions

The HelTH branded food composition database was restricted and expanded to include allergen data. This allowed for the first time to present information and analyses on the presence of allergens and the nutritional composition of products bearing allergen-free claims in 3859 branded food products in Greece. Approximately three quarters of the food products analysed contain at least one allergen at their ingredient list and/or in a precautionary statement. This indicates the difficulty in identifying suitable food products for individuals living with allergies, a choice that is significantly restricted by the frequent use of precautionary statements. Related to the nutritional composition of products carrying allergen-free claims, in general, allergen-free claimed products are not nutritionally superior compared to their counterparts. The inclusion of allergens data in the first BFCD in Greece is an important milestone in the study of nutrition and health; this provides a first overview on allergens and permits us to examine changes in labelling practices and the uses of allergens as ingredients over time. It appears that additional strategies for allergen and precautionary allergen labelling regulations should be implemented to minimize and standardize the allergen declaration on branded food products and create a friendlier, to food-allergic populations, food environment.

## Figures and Tables

**Figure 1 nutrients-14-03421-f001:**
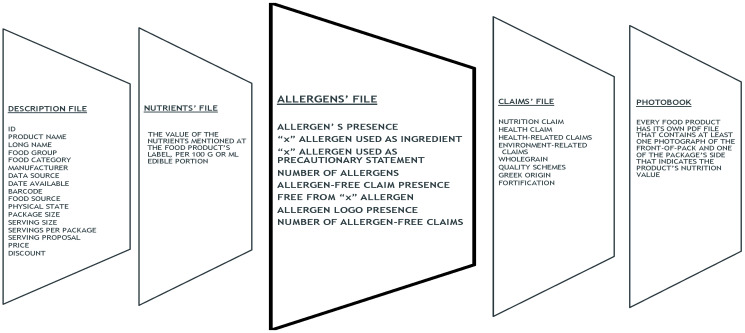
The new structure of the HelTH database. In the newly introduced allergen’s file, “x allergen” refers to any of the 14 allergens mentioned at the EU regulation [28] (cereals containing gluten (namely wheat, rye, barley, spelt, kamut, or their hybridized strains), crustaceans, eggs, fish, peanuts, soybeans (including soya lecithin), milk (including lactose), nuts (namely almonds, hazelnuts, walnuts, cashews, pecan nuts, Brazil nuts, pistachio nuts, macadamia or Queensland nuts), celery, mustard, sesame seeds, sulphite, lupin, and molluscs, including products thereof). Adapted from Katidi et al. [3].

**Figure 2 nutrients-14-03421-f002:**
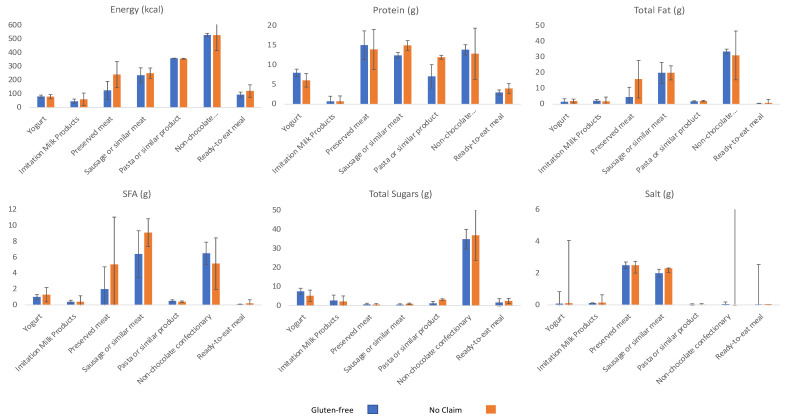
Comparison of the nutritional composition of products carrying a gluten-free claim to the corresponding ones that do not, in food subcategories with at least 5 products with and without a gluten-free claim.

**Table 1 nutrients-14-03421-t001:** Number of food products, per category and subcategory, in the HelTH database that presented on their labels information related to allergens (presence or absence).

Food Categories	Food Subcategories	Number of Products
Milk, milk product, or milk substitute		690
	Cream	40
	Milk	172
	Yogurts	170
	Cheese	213
	Milk imitation products	49
	Frozen dairy desserts	46
Egg or egg product		35
	Egg fresh or processed	35
Meat or meat product		136
	Canned meat	80
	Sausage or similar products	35
	Meat dish	21
Seafood or seafood product		75
	Seafood product	75
Fat or oil		81
	Vegetable fat or oil	8
	Margarine or mixed origin fat	39
	Butter or animal fat	34
Grain or grain product		1055
	Cereal or cereal milling products	51
	Rice or similar product	97
	Pasta or similar product	201
	Breakfast cereals	149
	Bread or similar product	212
	Fine bakery ware	259
	Savoury cereal dish	86
Nuts, seeds, or kernel		128
	Nuts	66
	Seeds or Kernel	35
	Nuts or seeds products	27
Vegetable or vegetable product		244
	Vegetables (excluding potato)	172
	Starchy root or potato	21
	Pulse or pulse product	51
Fruit or fruit product		43
	Processed fruit product	43
Sugar or sugar product		404
	Sugar, honey, or syrup	46
	Marmalade	83
	Sugary products (no chocolate)	68
	Chocolate	207
Beverages (non-milk)		448
	Juice or nectar	165
	Non-alcoholic beverage	283
Miscellaneous food products		442
	Spices or condiments	282
	Prepared food product	160
Meals		78
	Ready-to-eat meal	38
	Frozen, semi-ready meal	40
Number of food products		3859

**Table 2 nutrients-14-03421-t002:** Prevalence of products with allergen declaration for each 1 of the 14 allergens.

Allergen Category	Allergen Declaration in the Ingredient List	Allergen Declaration Only in a Precautionary Statement	Absence of Any Allergen Declaration
	*n* (%)	*n* (%)	*n* (%)
Milk (including lactose)	1498 (38.8)	469 (12.2)	1892 (49.0)
Cereals including gluten	1260 (32.7)	352 (9.1)	2247 (58.2)
Soybean	673 (17.4)	423 (11.0)	2763 (71.6)
Eggs	304 (7.9)	507 (13.1)	3048 (79.0)
Nuts	251 (6.5)	706 (18.3)	2902 (75.2)
Sesame	122 (3.2)	485 (12.6)	3252 (84.3)
Mustard	120 (3.1)	225 (5.8)	3514 (91.1)
Sulphite	105 (2.7)	98 (2.5)	3656 (94.7)
Fish	86 (2.2)	80 (2.1)	3693 (95.7)
Celery	85 (2.2)	244 (6.3)	3530 (91.5)
Peanuts	63 (1.6)	349 (9.1)	3447 (89.3)
Molluscs	10 (0.3)	16 (0.4)	3834 (99.4)
Crustaceans	5 (0.1)	17 (0.4)	3837 (99.4)
Lupin	4 (0.1)	48 (1.2)	3809 (98.7)
At least one allergen declared	2640 (68.4)	1501 (38.9)	
No declaration of any allergen			1052 (27.3)

**Table 3 nutrients-14-03421-t003:** Prevalence of the most commonly declared allergens in the ingredient list among the 3859 products in HelTH per food subcategory.

Food Subcategories	Top 7 Allergens Declared in the Ingredients List (*n* (%))						
	Milk	Cereals	Soybean	Eggs	Nuts	Sesame	Mustard
Cream	40 (100)	0 (0)	2 (5)	0 (0)	0 (0)	0 (0)	0 (0)
Milk	172 (100)	0 (0)	3 (2)	0 (0)	0 (0)	0 (0)	0 (0)
Yogurts	170 (100)	23 (14)	6 (4)	3 (2)	8 (5)	10 (6)	0 (0)
Cheese	213 (100)	3 (1)	0 (0)	1 (0)	0 (0)	0 (0)	0 (0)
Milk imitation products	2 (4)	2 (4)	19 (39)	0 (0)	17 (35)	0 (0)	0 (0)
Frozen dairy desserts	44 (96)	9 (20)	25 (54)	9 (20)	16 (35)	0 (0)	0 (0)
Eggs	0 (0)	0 (0)	0 (0)	35 (100)	0 (0)	0 (0)	0 (0)
Canned meat	33 (41)	4 (5)	22 (28)	0 (0)	0 (0)	0 (0)	8 (10)
Sausage or similar products	13 (37)	2 (6)	11 (31)	0 (0)	0 (0)	0 (0)	4 (11)
Meat dish	11 (52)	17 (81)	8 (38)	4 (19)	0 (0)	0 (0)	2 (10)
Seafood products	1 (1)	8 (11)	0 (0)	0 (0)	0 (0)	0 (0)	1 (1)
Vegetable fat or oil	3 (38)	0 (0)	0 (0)	0 (0)	0 (0)	0 (0)	0 (0)
Margarine or mixed origin fat	21 (54)	0 (0)	1 (3)	0 (0)	0 (0)	0 (0)	0 (0)
Butter or animal fat	34 (100)	0 (0)	0 (0)	0 (0)	0 (0)	0 (0)	0 (0)
Cereal or cereal milling products	11 (22)	51 (100)	12 (24)	7 (14)	0 (0)	0 (0)	0 (0)
Rice or similar product	7 (7)	27 (28)	14 (14)	5 (5)	0 (0)	0 (0)	3 (3)
Pasta or similar product	30 (15)	194 (97)	1 (0)	29 (14)	0 (0)	0 (0)	0 (0)
Breakfast cereals	63 (42)	137 (92)	65 (44)	0 (0)	26 (17)	2 (2)	0 (0)
Bread or similar product	30 (14)	206 (97)	32 (15)	9 (4)	1 (0)	39 (18)	3 (1)
Fine bakery ware	182 (70)	257 (99)	154 (59)	80 (31)	39 (15)	6 (2)	0 (0)
Savoury cereal dish	82 (95)	86 (100)	21 (24)	31 (36)	0 (0)	4 (5)	2 (2)
Nuts	2 (3)	19 (29)	4 (6)	0 (0)	40 (61)	1 (2)	0 (0)
Seeds or Kernel	1 (3)	0 (0)	0 (0)	0 (0)	0 (0)	0 (0)	0 (0)
Nuts or seeds products	1 (4)	0 (0)	6 (22)	0 (0)	2 (7)	21 (78)	0 (0)
Vegetables (excluding potato)	4 (2)	1 (1)	0 (0)	0 (0)	0 (0)	0 (0)	1 (1)
Starchy root or potato	2 (10)	0 (0)	0 (0)	0 (0)	0 (0)	0 (0)	0 (0)
Pulse or pulse product	0 (0)	1 (2)	0 (0)	0 (0)	0 (0)	0 (0)	0 (0)
Processed fruit product	0 (0)	0 (0)	0 (0)	0 (0)	0 (0)	0 (0)	0 (0)
Sugar, honey, or syrup	0 (0)	0 (0)	0 (0)	0 (0)	0 (0)	0 (0)	0 (0)
Marmalade	1 (1)	2 (2)	0 (0)	1 (1)	1 (1)	0 (0)	0 (0)
Sugary products (no chocolate)	1 (1)	0 (0)	2 (3)	3 (4)	14 (21)	38 (56)	0 (0)
Chocolate	159 (77)	49 (24)	182 (88)	1 (0)	75 (36)	1 (0)	0 (0)
Juice or nectar	0 (0)	0 (0)	0 (0)	0 (0)	0 (0)	0 (0)	0 (0)
Non-alcoholic beverage	5 (2)	2 (1)	10 (4)	0 (0)	0 (0)	0 (0)	0 (0)
Spices or condiments	60 (21)	64 (23)	33 (12)	43 (15)	7 (2)	2 (1)	60 (21)
Prepared food product	75 (47)	55 (34)	21 (13)	26 (16)	1 (1)	5 (3)	24 (15)
Ready-to-eat meal	19 (50)	25 (66)	11 (29)	15 (39)	3 (8)	2 (5)	12 (32)
Frozen, semi-ready meal	6 (15)	16 (40)	8 (20)	2 (5)	1 (3)	1 (3)	0 (0)
Total	1498 (39)	1260 (33)	673 (17)	304 (8)	251 (7)	132 (3)	120 (3)

**Table 4 nutrients-14-03421-t004:** Prevalence of the most commonly declared allergens in a precautionary statement among the 3859 products in HelTH per food subcategory.

Food Subcategories	Top 7 Allergens Declared in the Ingredients List (*n* (%))						
	Milk	Cereals	Soybeans	Eggs	Nuts	Sesame	Peanuts
Cream	0 (0)	0 (0)	0 (0)	0 (0)	0 (0)	0 (0)	0 (0)
Milk	0 (0)	0 (0)	0 (0)	0 (0)	0 (0)	0 (0)	0 (0)
Yogurts	0 (0)	8 (5)	4 (2)	7 (4)	12 (7)	10 (6)	7 (4)
Cheese	0 (0)	2 (1)	3 (1)	6 (3)	0 (0)	0 (0)	0 (0)
Milk imitation products	0 (0)	0 (0)	1 (2)	0 (0)	6 (12)	0 (0)	0 (0)
Frozen dairy desserts	0 (0)	16 (35)	11 (24)	18 (39)	22 (48)	4 (9)	18 (39)
Eggs	0 (0)	0 (0)	0 (0)	0 (0)	0 (0)	0 (0)	0 (0)
Canned meat	20 (25)	12 (15)	34 (43)	29 (36)	42 (53)	1 (1)	6 (8)
Sausage or similar products	8 (23)	4 (11)	7 (20)	13 (37)	11 (31)	0 (0)	4 (11)
Meat dish	8 (38)	2 (10)	0 (0)	3 (14)	3 (14)	0 (0)	0 (0)
Seafood products	1 (1)	1 (1)	1 (1)	1 (1)	0 (0)	0 (0)	0 (0)
Vegetable fat or oil	0 (0)	0 (0)	0 (0)	0 (0)	0 (0)	0 (0)	0 (0)
Margarine or mixed origin fat	12 (31)	0 (0)	0 (0)	0 (0)	0 (0)	0 (0)	0 (0)
Butter or animal fat	0 (0)	0 (0)	0 (0)	0 (0)	0 (0)	0 (0)	0 (0)
Cereal or cereal milling products	25 (25)	0 (0)	15 (29)	28 (55)	15 (29)	21 (41)	4 (8)
Rice or similar product	12 (12)	4 (4)	8 (8)	17 (18)	3 (3)	16 (16)	8 (8)
Pasta or similar product	12 (6)	0 (0)	73 (36)	47 (23)	1 (0)	4 (2)	1 (0)
Breakfast cereals	55 (37)	3 (2)	29 (19)	3 (2)	87 (58)	25 (17)	43 (29)
Bread or similar product	87 (41)	2 (1)	38 (18)	79 (37)	46 (22)	100 (47)	4 (2)
Fine bakery ware	46 (18)	0 (0)	56 (22)	104 (40)	163 (63)	131 (51)	65 (25)
Savoury cereal dish	3 (3)	0 (0)	24 (28)	35 (41)	36 (42)	61 (71)	1 (1)
Nuts	2 (3)	38 (58)	2 (3)	0 (0)	24 (36)	43 (65)	40 (61)
Seeds or Kernel	0 (0)	0 (0)	0 (0)	0 (0)	2 (6)	0 (0)	0 (0)
Nuts or seeds products	2 (2)	8 (30)	0 (0)	0 (0)	3 (11)	4 (15)	0 (0)
Vegetables (excluding potato)	0 (0)	0 (0)	0 (0)	0 (0)	1 (1)	0 (0)	0 (0)
Starchy root or potato	8 (38)	6 (29)	0 (0)	3 (14)	0 (0)	0 (0)	0 (0)
Pulse or pulse product	0 (0)	1 (2)	0 (0)	0 (0)	0 (0)	0 (0)	0 (0)
Processed fruit product	0 (0)	4 (9)	0 (0)	0 (0)	4 (9)	2 (5)	4 (9)
Sugar, honey, or syrup	0 (0)	0 (0)	0 (0)	0 (0)	0 (0)	0 (0)	0 (0)
Marmalade	2 (2)	9 (11)	0 (0)	0 (0)	0 (0)	0 (0)	0 (0)
Sugary products (no chocolate)	2 (3)	1 (1)	1 (1)	2 (3)	31 (46)	4 (6)	32 (47)
Chocolate	48 (23)	112 (54)	3 (1)	26 (13)	121 (58)	12 (6)	67 (32)
Juice or nectar	0 (0)	0 (0)	0 (0)	0 (0)	0 (0)	0 (0)	0 (0)
Non-alcoholic beverage	10 (4)	2 (1)	3 (1)	0 (0)	3 (1)	1 (0)	1 (0)
Spices or condiments	50 (18)	53 (19)	47 (17)	54 (19)	46 (16)	16 (6)	22 (8)
Prepared food product	46 (29)	56 (35)	51 (32)	20 (13)	18 (11)	15 (9)	16 (10)
Ready-to-eat meal	7 (18)	6 (16)	11 (29)	7 (18)	6 (16)	10 (26)	6 (16)
Frozen, semi-ready meal	3 (8)	1 (3)	0 (0)	5 (13)	0 (0)	5 (13)	0 (0)
Total	469 (12)	351 (9)	422 (11)	507 (13)	706 (18)	485 (13)	349 (9)

**Table 5 nutrients-14-03421-t005:** Prevalence of the three most commonly used allergen-free claims per food subcategories that include at least one allergen-free claim.

Food Subcategory	Gluten *n* (%)	Milk *n* (%)	Soy *n* (%)	At Least One Claim *n* (%)
Cream	2 (5)	0 (0)	0 (0)	2 (5)
Milk	2 (1)	6 (3)	0 (0)	8 (5)
Yogurts	9 (5)	4 (2)	0 (0)	13 (8)
Cheese	5 (2)	2 (1)	0 (0)	7 (3)
Milk imitation products	23 (47)	35 (71)	5 (10)	35 (71)
Frozen dairy desserts	2 (4)	2 (4)	0 (0)	4 (9)
Canned meat	39 (49)	3 (4)	0 (0)	39 (49)
Sausage or similar products	13 (37)	0 (0)	0 (0)	13 (37)
Seafood products	1 (1)	0 (0)	0 (0)	1 (1)
Butter or animal fat	0 (0)	1 (3)	0 (0)	1 (3)
Rice or similar product	1 (1)	0 (0)	0 (0)	1 (1)
Pasta or similar product	5 (2)	0 (0)	0 (0)	5 (2)
Bread or similar product	4 (2)	2 (1)	0 (0)	7 (3)
Bakery products	2 (1)	1 (0)	0 (0)	2 (1)
Nuts or seeds products	4 (15)	0 (0)	0 (0)	4 (15)
Starchy root or potato	4 (19)	0 (0)	0 (0)	4 (19)
Sugary products (no chocolate)	8 (12)	0 (0)	0 (0)	8 (12)
Chocolate	17 (8)	1 (0)	0 (0)	18 (9)
Juice or nectar	4 (2)	4 (2)	0 (0)	4 (2)
Non-alcoholic beverage	6 (2)	0 (0)	0 (0)	6 (2)
Spices or condiments	8 (3)	0 (0)	0 (0)	8 (3)
Prepared food product	14 (9)	1 (1)	1 (0.6)	14 (9)
Frozen, semi-ready meal	5 (13)	0 (0)	0 (0)	5 (13)
Total	178 (4.6)	62 (1.6)	6 (0.2)	206 (5.3)

## Data Availability

HelTH is available at https://www.eurofir.org/our-tools/foodexplorer/ (accessed on 14 August 2022).

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
