# Peer review of "The Expansion of the Hellenic Food Thesaurus; Allergens Labelling and Allergens-Free Claims on Greek Branded Food Products"

_nutrients, 2022, doi:10.3390/nu14163421_

Round 1

Reviewer 1 Report

In the present manuscript, Katidi et al. conduct an analysis of the allergen labelling and precautionary labelling of around 4000 products in Greece. This information is embedded in the Hellenic Food Thesaurus that is accessible online. The authors suggest that this information might help consumers in their decision making and provide information to the regulatory authorities. Overall, the study is well conducted and thorough, and mostly confirms previously published studies from France and Latin America.

Main concerns:

·         When attempting to retrieve information from Hellenic Food Thesaurus, I found it not so easy to do so. The website is not very intuitive and I did not manage to retrieve a single food product (probably because I didn’t understand its functioning correctly). Perhaps this is something to continue working on (e.g. the French oQuali website is much more intuitive).

·         The study lacks a bit of originality. Although necessary to conduct this type of analyses in multiple countries, the study largely copies previously published studies from other databases. An original point-of-view, a small addition, as compared to the other studies would be appreciated and is, I find, necessary for publishing in a journal as Nutrients.

·         I feel it should be possible to make some of the information available in a more graphical way, to make the article of interest and accessible to a larger public of scientists (instead or in addition to the tables presented in the article).

Minor concerns/comments:

·         Some small English grammatical errors (but overall well-written)

·         Do and if so how do the authors plan to integrate this database with the other local databases and how do they plan to augment the database over time?

Author Response

Point 1:     When attempting to retrieve information from Hellenic Food Thesaurus, I found it not so easy to do so. The website is not very intuitive, and I did not manage to retrieve a single food product (probably because I didn’t understand its functioning correctly). Perhaps this is something to continue working on (e.g. the French oQuali website is much more intuitive).

Response, Point 1:  We thank the reviewer for this comment. The HelTH website provided at first submission is at present under re-consturction. Instead the HelTH database is available at the EuroFIR’s website, https://www.eurofir.org/our-tools/foodexplorer/  and accessible at a membership fee.  The data availability statement is now revised (Lines 559, 560)

Point 2: The study lacks a bit of originality. Although necessary to conduct this type of analyses in multiple countries, the study largely copies previously published studies from other databases. An original point-of-view, a small addition, as compared to the other studies would be appreciated and is, I find, necessary for publishing in a journal as Nutrients.

Response, Point 2:  The study presented in the manuscript reflects largerly the need to fully describe the foodscape in relation to allergens and related claims; similar studies that inspired this work are cited.  To further promote the field, as suggested by the reviewer, additional analyses are now provided in the revised manuscript.  In these analyses we compare the nutritional composition of allergen-free claimed products to their corresponding counterparts with the objective to understand nutritional characteristics of products that bear allergen related claims (Lines 351-373).

Point 3:   I feel it should be possible to make some of the information available in a more graphical way, to make the article of interest and accessible to a larger public of scientists (instead or in addition to the tables presented in the article).

Response, Point 3: Six diagrams presenting the comparison of the content of energy and macronutrients between gluten-free products and their equivalents, per food subcategory, were added as a figure at the revised manuscript (Figure 1, pg 14).

Reviewer 2 Report

This manuscript showed us detailed information and analysis of the prevalence of allergens in the ingredient lists and/or precautionary statements and allergen-free statements on the Greek branded food products. Moreover, the HelTH branded food database was expanded by registering information on 14 allergens mentioned in the EU regulation. The research content of the manuscript simply filled the gap in the branded food database regarding allergen labeling data, which were extensive but not in-depth enough. The data analysis was too simple and did not uncover more valuable information. Furthermore, the issues found in recording allergen information in branded foods mentioned in the discussion section should be given a specific data analysis instead of just a simple description. 

Author Response

Point 1:     This manuscript showed us detailed information and analysis of the prevalence of allergens in the ingredient lists and/or precautionary statements and allergen-free statements on the Greek branded food products. Moreover, the HelTH branded food database was expanded by registering information on 14 allergens mentioned in the EU regulation. The research content of the manuscript simply filled the gap in the branded food database regarding allergen labeling data, which were extensive but not in-depth enough. The data analysis was too simple and did not uncover more valuable information.

Response, Point 1: We thank the reviewer for this comment. The study presented in the manuscript reflects largerly the need to fully describe the foodscape in relation to allergens and related claims. To further promote the field and reveal more in-depth information about branded foods bearing such claims related to allergens, additional analyses are now provided in the revised manuscript.  In these analyses we compare the nutritional composition of allergen-free claimed products to their corresponding counterparts with the objective to understand nutritional characteristics of products that bear allergen related claims (Lines 351-373).

Point 2: Furthermore, the issues found in recording allergen information in branded foods mentioned in the discussion section should be given a specific data analysis instead of just a simple description.

Response, Point 2: More detailed information to quantify the range of specific issues found in recording allergen information are now added to the revised manuscript (Lines. 459-464, 475-478).

Round 2

Reviewer 2 Report

The revised version is well respond to my comments.